# Cooperative Holistic Scene Understanding: Unifying 3D Object, Layout, and Camera Pose Estimation

**Siyuan Huang** [1]
huangsiyuan@ucla.edu

**Siyuan Qi** [2]
syqi@cs.ucla.edu

**Yinxue Xiao** [2]
yinxuex@ucla.edu

**Yixin Zhu** [1]
yixin.zhu@ucla.edu

**Ying Nian Wu** [1]
ywu@stat.ucla.edu

**Song-Chun Zhu** [1,2]
sczhu@stat.ucla.edu

[1] Dept. of Statistics, UCLA    [2] Dept. of Computer Science, UCLA

## Abstract

Holistic 3D indoor scene understanding refers to jointly recovering the i) object bounding boxes, ii) room layout, and iii) camera pose, all in 3D. The existing methods either are ineffective or only tackle the problem partially. In this paper, we propose an end-to-end model that *simultaneously* solves all three tasks in *real-time* given only a single RGB image. The essence of the proposed method is to improve the prediction by i) *parametrizing* the targets (*e.g.*, 3D boxes) instead of directly estimating the targets, and ii) *cooperative training* across different modules in contrast to training these modules individually. Specifically, we parametrize the 3D object bounding boxes by the predictions from several modules, *i.e.*, 3D camera pose and object attributes. The proposed method provides two major advantages: i) The parametrization helps maintain the consistency between the 2D image and the 3D world, thus largely reducing the prediction variances in 3D coordinates. ii) Constraints can be imposed on the parametrization to train different modules simultaneously. We call these constraints "cooperative losses" as they enable the joint training and inference. We employ three cooperative losses for 3D bounding boxes, 2D projections, and physical constraints to estimate a *geometrically consistent* and *physically plausible* 3D scene. Experiments on the SUN RGB-D dataset shows that the proposed method significantly outperforms prior approaches on 3D object detection, 3D layout estimation, 3D camera pose estimation, and holistic scene understanding.

## 1 Introduction

Holistic 3D scene understanding from a single RGB image is a fundamental yet challenging computer vision problem, while humans are capable of performing such tasks effortlessly within 200 ms [Potter, 1975, 1976, Schyns and Oliva, 1994, Thorpe et al., 1996]. The primary difficulty of the holistic 3D scene understanding lies in the vast, but ambiguous 3D information attempted to recover from a single RGB image. Such estimation includes three essential tasks:

- The estimation of the 3D camera pose that captures the image. This component helps to maintain the *consistency* between the 2D image and the 3D world.
- The estimation of the 3D room layout. Combining with the estimated 3D camera pose, it recovers a *global* geometry.
- The estimation of the 3D bounding boxes for each object in the scene, recovering the *local* details.

Most current methods either are inefficient or only tackle the problem partially. Specifically,

- Traditional methods [Gupta et al., 2010, Zhao and Zhu, 2011, 2013, Choi et al., 2013, Schwing et al., 2013, Zhang et al., 2014, Izadinia et al., 2017, Huang et al., 2018] apply sampling or optimization methods to infer the geometry and semantics of indoor scenes. However, those methods are computationally expensive; it usually takes a long time to converge and could be easily trapped in

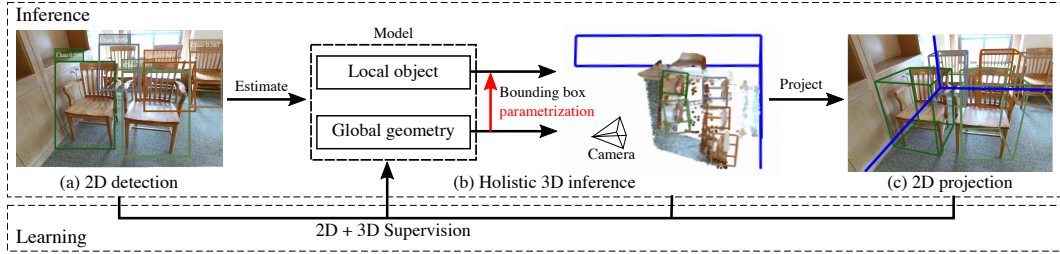

Figure 1: Overview of the proposed framework for cooperative holistic scene understanding. (a) We first detect 2D objects and generate their bounding boxes, given a single RGB image as the input, from which (b) we can estimate 3D object bounding boxes, 3D room layout, and 3D camera pose. The blue bounding box is the estimated 3D room layout. (c) We project 3D objects to the image plane with the learned camera pose, forcing the projection from the 3D estimation to be consistent with 2D estimation.

an unsatisfactory local minimum, especially for cluttered indoor environments. Thus both stability and scalability become issues.

• Recently, researchers attempt to tackle this problem using deep learning. The most straightforward way is to directly predict the desired targets (*e.g.*, 3D room layouts or 3D bounding boxes) by training the individual modules separately with isolated losses for each module. Thereby, the prior work [Mousavian et al., 2017, Lee et al., 2017, Kehl et al., 2017, Kundu et al., 2018, Zou et al., 2018, Liu et al., 2018] only focuses on the individual tasks or learn these tasks separately rather than jointly inferring all three tasks, or only considers the inherent relations without explicitly modeling the connections among them [Tulsiani et al., 2018].

• Another stream of approach takes both an RGB-D image and the camera pose as the input [Lin et al., 2013, Song and Xiao, 2014, 2016, Song et al., 2017, Deng and Latecki, 2017, Zou et al., 2017, Qi et al., 2018, Lahoud and Ghanem, 2017, Zhang et al., 2017a], which provides sufficient geometric information from the depth images, thereby relying less on the consistency among different modules.

In this paper, we aim to address the missing piece in the literature: to recover a *geometrically consistent* and *physically plausible* 3D scene and jointly solve all three tasks in an *efficient* and *cooperative* way, only from a single RGB image. Specifically, we tackle three important problems:

1. *2D-3D consistency*   A good solution to the aforementioned three tasks should maintain a high consistency between the 2D image plane and the 3D world coordinate. How should we design a method to achieve such consistency?

2. *Cooperation*   Psychological studies have shown that our biologic perception system is extremely good at rapid scene understanding [Schyns and Oliva, 1994], particularly utilizing the fusion of different visual cues [Landy et al., 1995, Jacobs, 2002]. Such findings support the necessities of cooperatively solving all the holistic scene tasks together. Can we devise an algorithm such that it can *cooperatively* solve these tasks, making different modules reinforce each other?

3. *Physically Plausible*   As humans, we excel in inferring the physical attributes and dynamics [Kubricht et al., 2017]. Such a deep understanding of the physical environment is imperative, especially for an interactive agent (*e.g.*, a robot) to navigate the environment or collaborate with a human agent. How can the model estimate a 3D scene in a physically plausible fashion, or at least have some sense of physics?

To address these issues, we propose a novel parametrization of the 3D bounding box as well as a set of cooperative losses. Specifically, we parametrize the 3D boxes by the predicted camera pose and object attributes from individual modules. Hence, we can construct the 3D boxes starting from the 2D box centers to maintain a 2D-3D consistency, rather than predicting 3D coordinates directly or assuming the camera pose is given, which loses the 2D-3D consistency.

Cooperative losses are further imposed on the parametrization in addition to the direct losses to enable the joint training of all the individual modules. Specifically, we employ three cooperative losses on the parametrization to constrain the 3D bounding boxes, projected 2D bounding boxes, and physical plausibility, respectively:

• The 3D bounding box loss encourages accurate 3D estimation.

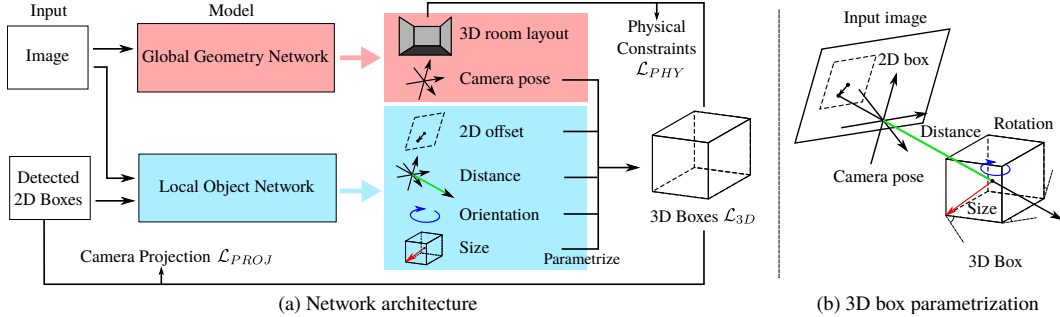

Figure 2: Illustration of (a) network architecture and (b) parametrization of 3D object bounding box.

- The differentiable 2D projection loss measures the consistency between 3D and 2D bounding boxes, which permits our networks to learn the 3D structures with only 2D annotations (*i.e.*, no 3D annotations are required). In fact, we can directly supervise the learning process with 2D objects annotations using the common sense of the object sizes.
- The physical plausibility loss penalizes the intersection between the reconstructed 3D object boxes and the 3D room layout, which prompts the networks to yield a physically plausible estimation.

Figure 1 shows the proposed framework for cooperative holistic scene understanding. Our method starts with the detection of 2D object bounding boxes from a single RGB image. Two branches of convolutional neural networks are employed to learn the 3D scene from both the image and 2D boxes: i) The *global geometry network* (GGN) learns the global geometry of the scene, predicting both the 3D room layout and the camera pose. ii) The *local object network* (LON) learns the object attributes, estimating the object pose, size, distance between the 3D box center and camera center, and the 2D offset from the 2D box center to the projected 3D box center on the image plane. The details are discussed in Section 2. By combining the camera pose from the GGN and object attributes from the LON, we can parametrize 3D bounding boxes, which grants jointly learning of both GGN and LON with 2D and 3D supervisions.

Another benefit of the proposed parametrization is improving the training stability by reducing the variance of the 3D boxes prediction, due to that i) the estimated 2D offset has relatively low variance, and ii) we adopt a hybrid of classification and regression method to estimate the variables of large variances, inspired by [Ren et al., 2015, Mousavian et al., 2017, Qi et al., 2018].

We evaluate our method on SUN RGB-D Dataset [Song et al., 2015]. The proposed method outperforms previous methods on four tasks, including 3D layout estimation, 3D object detection, 3D camera pose estimation, and holistic scene understanding. Our experiments demonstrate that a cooperative method performing holistic scene understanding tasks can significantly outperform existing methods tackling each task in isolation, further indicating the necessity of joint training.

Our contributions are four-fold. i) We formulate an end-to-end model for 3D holistic scene understanding tasks. The essence of the proposed model is to cooperatively estimate 3D room layout, 3D camera pose, and 3D object bounding boxes. ii) We propose a novel parametrization of the 3D bounding boxes and integrate physical constraint, enabling the cooperative training of these tasks. iii) We bridge the gap between the 2D image plane and the 3D world by introducing a differentiable objective function between the 2D and 3D bounding boxes. iv) Our method significantly outperforms the state-of-the-art methods and runs in real-time.

## 2 Method

In this section, we describe the parametrization of the 3D bounding boxes and the neural networks designed for the 3D holistic scene understanding. The proposed model consists of two networks, shown in Figure 2: a *global geometric network* (GGN) that estimates the 3D room layout and camera pose, and a *local object network* (LON) that infers the attributes of each object. Based on these two networks, we further formulate differentiable loss functions to train the two networks cooperatively.

### 2.1 Parametrization

**3D Objects**  We use the 3D bounding box $X^W \in \mathbb{R}^{3 \times 8}$ as the representation of the estimated 3D object in the world coordinate. The 3D bounding box is described by its 3D center $C^W \in \mathbb{R}^3$, size

$S^W \in \mathbb{R}^3$, and orientation $R(\theta^W) \in \mathbb{R}^{3\times3}$: $X^W = h(C^W, R(\theta^W), S)$, where $\theta$ is the heading angle along the up-axis, and $h(\cdot)$ is the function that composes the 3D bounding box.

Without any depth information, estimating 3D object center $C^W$ directly from the 2D image may result in a large variance of the 3D bounding box estimation. To alleviate this issue and bridge the gap between 2D and 3D object bounding boxes, we parametrize the 3D center $C^W$ by its corresponding 2D bounding box center $C^I \in \mathbb{R}^2$ on the image plane, distance $D$ between the camera center and the 3D object center, the camera intrinsic parameter $K \in \mathbb{R}^{3\times3}$, and the camera extrinsic parameters $R(\phi, \psi) \in \mathbb{R}^{3\times3}$ and $T \in \mathbb{R}^3$, where $\phi$ and $\psi$ are the camera rotation angles. As illustrated in Figure 2(b), since each 2D bounding box and its corresponding 3D bounding box are both manually annotated, there is always an offset $\delta^I \in \mathbb{R}^2$ between the 2D box center and the projection of 3D box center. Therefore, the 3D object center $C^W$ can be computed as

$$C^W = T + DR(\phi, \psi)^{-1} \frac{K^{-1}\left[C^I + \delta^I, 1\right]^T}{\left\|K^{-1}\left[C^I + \delta^I, 1\right]^T\right\|}. \tag{1}$$

Since $T$ becomes $\vec{0}$ when the data is captured from the first-person view, the above equation could be written as $C^W = p(C^I, \delta^I, D, \phi, \psi, K)$, where $p$ is a differentiable projection function.

In this way, the parametrization of the 3D object bounding box unites the 3D object center $C^W$ and 2D object center $C^I$, which helps maintain the 2D-3D consistency and reduces the variance of the 3D bounding box estimation. Moreover, it integrates both object attributes and camera pose, promoting the cooperative training of the two networks.

**3D Room Layout**　Similar to 3D objects, we parametrize 3D room layout in the world coordinate as a 3D bounding box $X^L \in \mathbb{R}^{3\times8}$, which is represented by its 3D center $C^L \in \mathbb{R}^3$, size $S^L \in \mathbb{R}^3$, and orientation $R(\theta^L) \in \mathbb{R}^{3\times3}$, where $\theta^L$ is the rotation angle. In this paper, we estimate the room layout center by predicting the offset from the pre-computed average layout center.

## 2.2　Direct Estimations

As shown in Figure 2(a), the *global geometry network* (GGN) takes a single RGB image as the input, and predicts both 3D room layout and 3D camera pose. Such design is driven by the fact that the estimations of both the 3D room layout and 3D camera pose rely on low-level global geometric features. Specifically, GGN estimates the center $C^L$, size $S^L$, and the heading angle $\theta^L$ of the 3D room layout, as well as the two rotation angles $\phi$ and $\psi$ for predicting the camera pose.

Meanwhile, the *local object network* (LON) takes 2D image patches as the input. For each object, LON estimates object attributes including distance $D$, size $S^W$, heading angle $\theta^W$, and the 2D offsets $\delta^I$ between the 2D box center and the projection of the 3D box center.

Direct estimations are supervised by two losses $\mathcal{L}_{\text{GGN}}$ and $\mathcal{L}_{\text{LON}}$. Specifically, $\mathcal{L}_{\text{GGN}}$ is defined as

$$\mathcal{L}_{\text{GGN}} = \mathcal{L}_{\phi} + \mathcal{L}_{\psi} + \mathcal{L}_{C^L} + \mathcal{L}_{S^L} + \mathcal{L}_{\theta^L}, \tag{2}$$

and $\mathcal{L}_{\text{LON}}$ is defined as

$$\mathcal{L}_{\text{LON}} = \frac{1}{N} \sum_{j=1}^{N} (\mathcal{L}_{D_j} + \mathcal{L}_{\delta_j^I} + \mathcal{L}_{S_j^W} + \mathcal{L}_{\theta_j^W}), \tag{3}$$

where $N$ is the number of objects in the scene. In practice, directly regressing objects' attributes (*e.g.*, heading angle) may result in a large error. Inspired by [Ren et al., 2015, Mousavian et al., 2017, Qi et al., 2018], we adopt a hybrid method of classification and regression to predict the sizes and heading angles. Specifically, we pre-define several size templates or equally split the space into a set of angle bins. Our model first classifies size and heading angles to those pre-defined categories, and then predicts residual errors within each category. For example, in the case of the rotation angle $\phi$, we define $\mathcal{L}_{\phi} = \mathcal{L}_{\phi-cls} + \mathcal{L}_{\phi-reg}$. Softmax is used for classification and smooth-L1 (Huber) loss is used for regression.

## 2.3　Cooperative Estimations

Psychological experiments have shown that human perception of the scene often relies on global information instead of local details, known as the gist of the scene [Oliva, 2005, Oliva and Torralba,

2006]. Furthermore, prior studies have demonstrated that human perceptions on specific tasks involve the cooperation from multiple visual cues, *e.g.*, on depth perception [Landy et al., 1995, Jacobs, 2002]. These crucial observations motivate the idea that the attributes and properties are naturally coupled and tightly bounded, thus should be estimated cooperatively, in which individual component would help to boost each other.

Using the parametrization described in Subsection 2.1, we hope to cooperatively optimize GGN and LON, simultaneously estimating 3D camera pose, 3D room layout, and 3D object bounding boxes, in the sense that the two networks enhance each other and cooperate to make the definitive estimation during the learning process. Specifically, we propose three cooperative losses which jointly provide supervisions and fuse 2D/3D information into a physically plausible estimation. Such cooperation improves the estimation accuracy of 3D bounding boxes, maintains the consistency between 2D and 3D, and generates a physically plausible scene. We further elaborate on these three aspects below.

**3D Bounding Box Loss**   As neither GGN or LON is directly optimized for the accuracy of the final estimation of the 3D bounding box, learning directly through GGN and LON is evidently not sufficient, thus requiring additional regularization. Ideally, the estimation of the object attributes and camera pose should be cooperatively optimized, as both contribute to the estimation of the 3D bounding box. To achieve this goal, we propose the 3D bounding box loss with respect to its 8 corners

$$\mathcal{L}_{\text{3D}} = \frac{1}{N} \sum_{j=1}^{N} \left\| h(C_j^W, R(\theta_j), S_j) - X_j^{W*} \right\|_2^2 , \tag{4}$$

where $X^{W*}$ is the ground truth 3D bounding boxes in the world coordinate. Qi et al. [2018] proposes a similar regularization in which the parametrization of 3D bounding boxes is different.

**2D Projection Loss**   In addition to the 3D parametrization of the 3D bounding boxes, we further impose an additional consistency as the 2D projection loss, which maintains the coherence between the 2D bounding boxes in the image plane and the 3D bounding boxes in the world coordinate. Specifically, we formulate the learning objective of the projection from 3D to 2D as

$$\mathcal{L}_{\text{PROJ}} = \frac{1}{N} \sum_{j=1}^{N} \left\| f(X_j^W, R, K) - X_j^{I*} \right\|_2^2 , \tag{5}$$

where $f(\cdot)$ denotes a differentiable projection function which projects a 3D bounding box to a 2D bounding box, and $X_j^{I*} \in \mathbb{R}^{2 \times 4}$ is the 2D object bounding box (either detected or the ground truth).

**Physical Loss**   In the physical world, 3D objects and room layout should not intersect with each other. To produce a physically plausible 3D estimation of a scene, we integrate the physical loss that penalizes the physical violations between 3D objects and 3D room layout

$$\mathcal{L}_{\text{PHY}} = \frac{1}{N} \sum_{j=1}^{N} \left( \text{ReLU}(\text{Max}(X_j^W) - \text{Max}(X^L)) + \text{ReLU}(\text{Min}(X^L) - \text{Min}(X_j^W)) \right), \tag{6}$$

where ReLU is the activate function, $\text{Max}(\cdot)$ / $\text{Min}(\cdot)$ takes a 3D bounding box as the input and outputs the max/min value along three world axes. By adding the physical constraint loss, the proposed model connects the 3D environments and the 3D objects, resulting in a more natural estimation of both 3D objects and 3D room layout.

To summarize, the total loss can be written as

$$\mathcal{L}_{\text{Total}} = \mathcal{L}_{\text{GGN}} + \mathcal{L}_{\text{LON}} + \lambda_{\text{COOP}} \left( \mathcal{L}_{\text{3D}} + \mathcal{L}_{\text{PROJ}} + \mathcal{L}_{\text{PHY}} \right), \tag{7}$$

where $\lambda_{\text{COOP}}$ is the trade-off parameter that balances the cooperative losses and the direct losses.

## 3   Implementation

Both the GGN and LON adopt ResNet-34 [He et al., 2016] architecture as the encoder, which encodes a 256x256 RGB image into a 2048-D feature vector. As each of the networks consists of multiple output channels, for each channel with an L-dimensional output, we stack two fully connected layers (2048-1024, 1024-L) on top of the encoder to make the prediction.

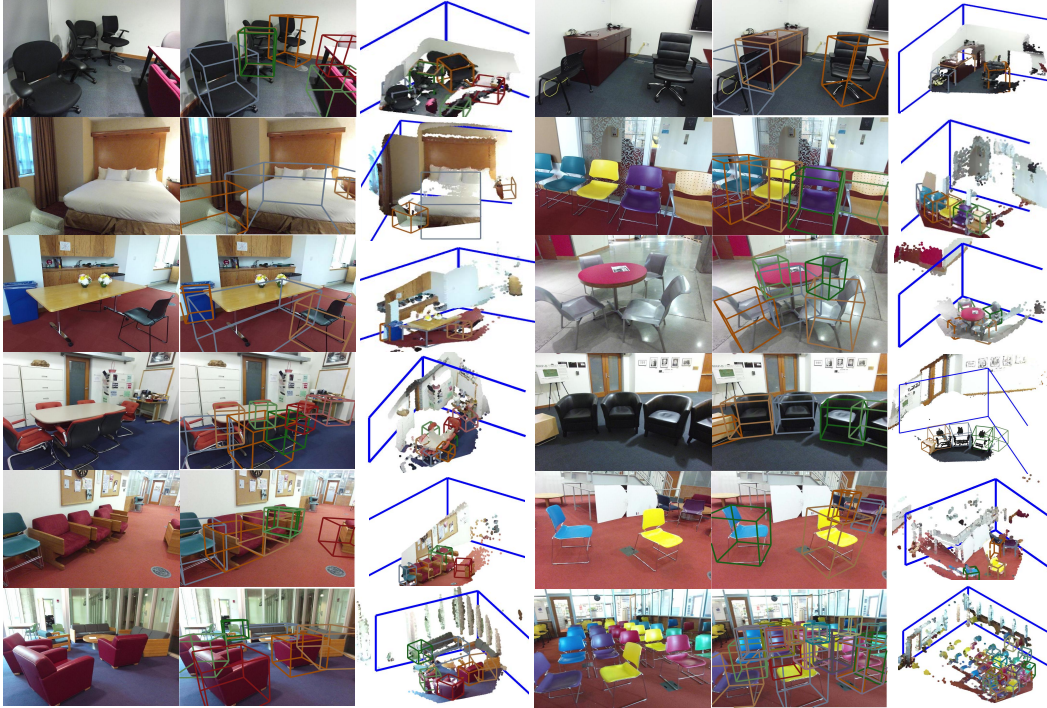

Figure 3: Qualitative results (top 50%). (Left) Original RGB images. (Middle) Results projected in 2D. (Right) Results in 3D. Note that the depth input is only used to visualize the 3D results.

We adopt a two-step training procedure. First, we fine-tune the 2D detector [Dai et al., 2017, Bodla et al., 2017] with 30 most common object categories to generate 2D bounding boxes. The 2D and 3D bounding box are matched to ensure each 2D bounding box has a corresponding 3D bounding box.

Second, we train two 3D estimation networks. To obtain good initial networks, both GGN and LON are first trained individually using the synthetic data (SUNCG dataset [Song et al., 2017]) with photo-realistically rendered images Zhang et al. [2017b]. We then fix six blocks of the encoders of GGN and LON, respectively, and fine-tune the two networks jointly on SUN RGBD dataset [Song et al., 2015].

To avoid over-fitting, a data augmentation procedure is performed by randomly flipping the images or randomly shifting the 2D bounding boxes with corresponding labels during the cooperative training. We use Adam [Kingma and Ba, 2015] for optimization with a batch size of 1 and a learning rate of 0.0001. In practice, we train the two networks cooperatively for ten epochs, which takes about 10 minutes for each epoch. We implement the proposed approach in PyTorch [Paszke et al., 2017].

## 4 Evaluation

We evaluate our model on SUN RGB-D dataset [Song et al., 2015], including 5050 test images and 10335 images in total. The SUN RGB-D dataset has 47 scene categories with high-quality 3D room layout, 3D camera pose, and 3D object bounding boxes annotations. It also provides benchmarks for various 3D scene understanding tasks. Here, we only use the RGB images as the input. Figure 3 shows some qualitative results. We discard the rooms with no detected 2D objects or invalid 3D room layout annotation, resulting in a total of 4783 training images and 4220 test images. More results can be found in the supplementary materials.

We evaluate our model on five tasks: i) 3D layout estimation, ii) 3D object detection, iii) 3D box estimation iv) 3D camera pose estimation, and v) holistic scene understanding, all with the test images across all scene categories. For each task, we compare our cooperatively trained model with the settings in which we train GGN and LON individually without the proposed parametrization of 3D object bounding box or cooperative losses. In the individual training setting, LON directly estimates the 3D object centers in the 3D world coordinate.

**3D Layout Estimation**  Since SUN RGB-D dataset provides the ground truth of 3D layout with arbitrary numbers of polygon corners, we parametrize each 3D room layout as a 3D bounding box by

Table 1: Comparison of 3D room layout estimation and holistic scene understanding on SUN RGB-D.

| Method | 3D Layout Estimation | Holistic Scene Understanding | | | |
|--------|:---:|:---:|:---:|:---:|:---:|
| | IoU | $P_g$ | $R_g$ | $R_r$ | IoU |
| 3DGP [Choi et al., 2013] | 19.2 | 2.1 | 0.7 | 0.6 | 13.9 |
| HoPR [Huang et al., 2018] | 54.9 | 37.7 | 23.0 | 18.3 | 40.7 |
| Ours (individual) | 55.4 | 36.8 | 22.4 | 20.1 | 39.6 |
| Ours (cooperative) | **56.9** | **49.3** | **29.7** | **28.5** | **42.9** |

Table 2: Comparisons of 3D object detection on SUN RGB-D.

| Method | bed | chair | sofa | table | desk | toilet | bin | sink | shelf | lamp | mAP |
|--------|:---:|:---:|:---:|:---:|:---:|:---:|:---:|:---:|:---:|:---:|:---:|
| Choi et al. [2013] | 5.62 | 2.31 | 3.24 | 1.23 | - | - | - | - | - | - | - |
| Huang et al. [2018] | 58.29 | 13.56 | 28.37 | 12.12 | 4.79 | 16.50 | 0.63 | 2.18 | 1.29 | 2.41 | 14.01 |
| Ours (individual) | 53.08 | 7.7 | 27.04 | 22.80 | 5.51 | 28.07 | 0.54 | 5.08 | 2.58 | 0.01 | 15.24 |
| Ours (cooperative) | **63.58** | **17.12** | **41.22** | **26.21** | **9.55** | **58.55** | **10.19** | **5.34** | **3.01** | **1.75** | **23.65** |

taking the output of the Manhattan Box baseline from [Song et al., 2015] with eight layout corners, which serves as the ground truth. We compare the estimation of the proposed model with three previous methods—3DGP [Choi et al., 2013], IM2CAD [Izadinia et al., 2017] and HoPR [Huang et al., 2018]. Following the evaluation protocol defined in [Song et al., 2015], we compute the average Intersection over Union (IoU) between the free space of the ground truth and the free space estimated by the proposed method. Table 1 shows our model outperforms HoPR by 2.0%. The results further show that there is an additional 1.5% performance improvement compared with individual training, demonstrating the efficacy of our method. Note that IM2CAD [Izadinia et al., 2017] manually selected 484 images from 794 test images of living rooms and bedrooms. For fair comparisons, we evaluate our method on the entire set of living room and bedrooms, outperforming IM2CAD by 2.1%.

**3D Object Detection** We evaluate our 3D object detection results using the metrics defined in [Song et al., 2015]. Specifically, the mean average precision (mAP) is computed using the 3D IoU between the predicted and the ground truth 3D bounding boxes. In the absence of depth, the threshold of IoU is adjusted from 0.25 (evaluation setting with depth image input) to 0.15 to determine whether two bounding boxes are overlapped. The 3D object detection results are reported in Table 2. We report 10 out of 30 object categories here, and the rest are reported in the supplementary materials. The results indicate our method outperforms HoPR by 9.64% on mAP and improves the individual training result by 8.41%. Compared with the model using individual training, the proposed cooperative model makes a significant improvement, especially on small objects such as bins and lamps. The accuracy of the estimation easily influences 3d detection of small objects; oftentimes, it is nearly impossible for prior approaches to detect. In contrast, benefiting from the parametrization method and 2D projection loss, the proposed cooperative model maintains the consistency between 3D and 2D, substantially reducing the estimation variance. Note that although IM2CAD also evaluates the 3D detection, they use a metric related to a specific distance threshold. For fair comparisons, we further conduct experiments on the subset of living rooms and bedrooms, using the same object categories with respect to this particular metric rather than an IoU threshold. We obtain an mAP of 78.8%, 4.2% higher than the results reported in IM2CAD.

**3D Box Estimation** The 3D object detection performance of our model is determined by both the 2D object detection and the 3D bounding box estimation. We first evaluate the accuracy of the 3D bounding box estimation, which reflects the ability to predict 3D boxes from 2D image patches. Instead of using mAP, 3D IoU is directly computed between the ground truth and the estimated 3D boxes for each object category. To evaluate the 2D-3D consistency, the estimated 3D boxes are projected back to 2D, and the 2D IoU is evaluated between the projected and detected 2D boxes. Results using the full model are reported in Table 3, which shows 3D estimation is still under satisfactory, despite the efforts to maintain a good 2D-3D consistency. The underlying reason for the gap between 3D and 2D performance is the increased estimation dimension. Another possible reason is due to the lack of context relations among objects. Results for all object categories can be found in the supplementary materials.

Table 3: 3D box estimation results on SUN RGB-D.

| | bed | chair | sofa | table | desk | toilet | bin | sink | shelf | lamp | mIoU |
|--------|:---:|:---:|:---:|:---:|:---:|:---:|:---:|:---:|:---:|:---:|:---:|
| IoU (3D) | 33.1 | 15.7 | 28.0 | 20.8 | 15.6 | 25.1 | 13.2 | 9.9 | 6.9 | 5.9 | 17.4 |
| IoU (2D) | 75.7 | 68.1 | 74.4 | 71.2 | 70.1 | 72.5 | 69.7 | 59.3 | 62.1 | 63.8 | 68.7 |

Table 4: Comparisons of 3D camera pose estimation on SUN RGB-D.

| Method | Mean Absolute Error (degree) | |
|---|---|---|
| | yaw | roll |
| Hedau et al. [2009] | 3.45 | 33.85 |
| Huang et al. [2018] | 3.12 | 7.60 |
| Ours (individual) | 2.48 | 4.56 |
| Ours (cooperative) | **2.19** | **3.28** |

**Camera Pose Estimation**  We evaluate the camera pose by computing the mean absolute error of yaw and roll between the model estimation and ground truth. As shown in Table 4, comparing with the traditional geometry-based method [Hedau et al., 2009] and previous learning-based method [Huang et al., 2018], the proposed cooperative model gains a significant improvement. It also improves the individual training performance with 0.29 degree on yaw and 1.28 degree on roll.

**Holistic Scene Understanding**  Per definition introduced in [Song et al., 2015], we further estimate the holistic 3D scene including 3D objects and 3D room layout on SUN RGB-D. Note that the holistic scene understanding task defined in [Song et al., 2015] misses 3D camera pose estimation compared to the definition in this paper, as the results are evaluated in the world coordinate.

Using the metric proposed in [Song et al., 2015], we evaluate the geometric precision $P_g$, the geometric recall $R_g$, and the semantic recall $R_r$ with the IoU threshold set to 0.15. We also evaluate the IoU between free space (3D voxels inside the room polygon but outside any object bounding box) of the ground truth and the estimation. Table 1 shows that we improve the previous approaches by a significant margin. Moreover, we further improve the individually trained results by 8.8% on geometric precision, 5.6% on geometric recall, 6.6% on semantic recall, and 3.7% on free space estimation. The performance gain of total scene understanding directly demonstrates that the effectiveness of the proposed parametrization method and cooperative learning process.

## 5 Discussion

In the experiment, the proposed method outperforms the state-of-the-art methods on four tasks. Moreover, our model runs at 2.5 fps (0.4s for 2D detection and 0.02s for 3D estimation) on a single Titan Xp GPU, while other models take significantly much more time; *e.g.*, [Izadinia et al., 2017] takes about 5 minutes to estimate one image. Here, we further analyze the effects of different components in the proposed cooperative model, hoping to shed some lights on how parametrization and cooperative training help the model using a set of ablative analysis.

### 5.1 Ablative Analysis

We compare four variants of our model with the full model trained using $\mathcal{L}_{\text{SUM}}$:

1. The model trained without the supervision on 3D object bounding box corners (w/o $\mathcal{L}_{\text{3D}}$, $S_1$).
2. The model trained without the 2D supervision (w/o $\mathcal{L}_{\text{PROJ}}$, $S_2$).
3. The model trained without the penalty of physical constraint (w/o $\mathcal{L}_{\text{PHY}}$, $S_3$).
4. The model trained in an unsupervised fashion where we only use 2D supervision to estimate the 3D bounding boxes (w/o $\mathcal{L}_{\text{3D}} + \mathcal{L}_{\text{GGN}} + \mathcal{L}_{\text{LON}}$, $S_4$).

Additionally, we compare two variants of training settings: i) the model trained directly on SUN RGB-D without pre-train ($S_5$), and ii) the model trained with 2D bounding boxes projected from ground truth 3D bounding boxes ($S_6$). We conduct the ablative analysis over all the test images on the task of holistic scene understanding. We also compare the 3D mIoU and 2D mIoU of 3D box estimation. Table 5 summarizes the quantitative results.

Table 5: The ablative analysis of the proposed cooperative model on SUN RGB-D. We evaluate holistic scene understanding, 3D mIoU and 2D mIoU of box estimation under different settings.

| Setting | $S_1$ | $S_2$ | $S_3$ | $S_4$ | $S_5$ | $S_6$ | Full |
|---|---|---|---|---|---|---|---|
| IoU | 42.8 | 42.0 | 41.7 | 35.9 | 40.2 | 43.0 | **43.3** |
| $P_g$ | 41.8 | **48.3** | 47.2 | 28.1 | 36.3 | 45.4 | 46.5 |
| $R_g$ | 25.3 | **30.1** | 27.5 | 17.1 | 22.1 | 29.7 | 28.0 |
| $R_r$ | 23.8 | **28.7** | 26.4 | 15.6 | 20.6 | 27.1 | 26.7 |
| 3D mIoU | 14.4 | **18.2** | 17.3 | 9.8 | 12.7 | 17.0 | 17.4 |
| 2D mIoU | 65.2 | 60.7 | 68.5 | 64.3 | 65.3 | 67.7 | **68.7** |

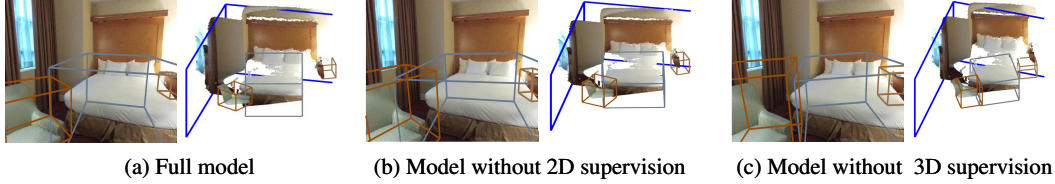

| (a) Full model | (b) Model without 2D supervision | (c) Model without 3D supervision |

Figure 4: Comparison with two variants of our model.

**Experiment S$_1$ and S$_3$**  Without the supervision on 3D object bounding box corners or physical constraint, the performance of all the tasks decreases since it removes the cooperation between the two networks.

**Experiment S$_2$**  The performance on the 3D detection is improved without the projection loss, while the 2D mIoU decreases by 8.0%. As shown in Figure 4(b), a possible reason is that the 2D-3D consistency $\mathcal{L}_{\text{PROJ}}$ may hurt the performance on 3D accuracy compared with directly using 3D supervision, while the 2D performance is largely improved thanks to the consistency.

**Experiment S$_4$**  The training entirely in an unsupervised fashion for 3D bounding box estimation would fail since each 2D pixel could correspond to an infinite number of 3D points. Therefore, we integrate some common sense into the unsupervised training by restricting the size of the object close to the average size. As shown in Figure 4(c), we can still estimate the 3D bounding box without 3D supervision quite well, although the orientations are usually not accurate.

**Experiment S$_5$ and S$_6$**  $S_5$ demonstrates the efficiency of using a large amount of synthetic training data, and $S_6$ indicates that we can gain almost the same performance even if there are no 2D bounding box annotations.

## 5.2   Related Work

**Single Image Scene Reconstruction**  Existing 3D scene reconstruction approaches fall into two streams. i) Generative approaches model the reconfigurable graph structures in generative probabilistic models [Zhao and Zhu, 2011, 2013, Choi et al., 2013, Lin et al., 2013, Guo and Hoiem, 2013, Zhang et al., 2014, Zou et al., 2017, Huang et al., 2018]. ii) Discriminative approaches [Izadinia et al., 2017, Tulsiani et al., 2018, Song et al., 2017] reconstruct the 3D scene using the representation of 3D bounding boxes or voxels through direct estimations. Generative approaches are better at modeling and inferring scenes with complex context, but they rely on sampling mechanisms and are always computational ineffective. Compared with prior discriminative approaches, our model focus on establishing cooperation among each scene module.

**Gap between 2D and 3D**  It is intuitive to constrain the 3D estimation to be consistent with 2D images. Previous research on 3D shape completion and 3D object reconstruction explores this idea by imposing differentiable 2D-3D constraints between the shape and silhouettes [Wu et al., 2016, Rezende et al., 2016, Yan et al., 2016, Tulsiani and Malik, 2015, Wu et al., 2017]. Mousavian et al. [2017] infers the 3D bounding boxes by matching the projected 2D corners in autonomous driving. In the proposed cooperative model, we introduce the parametrization of the 3D bounding box, together with a differentiable loss function to impose the consistency between 2D-3D bounding boxes for indoor scene understanding.

## 6   Conclusion

Using a single RGB image as the input, we propose an end-to-end model that recovers a 3D indoor scene in real-time, including the 3D room layout, camera pose, and object bounding boxes. A novel parametrization of 3D bounding boxes and a 2D projection loss are introduced to enforce the consistency between 2D and 3D. We also design differentiable cooperative losses which help to train two major modules cooperatively and efficiently. Our method shows significant improvements in various benchmarks while achieving high accuracy and efficiency.

**Acknowledgement:** The work reported herein was supported by DARPA XAI grant N66001-17-2-4029, ONR MURI grant N00014-16-1-2007, ARO grant W911NF-18-1-0296, and an NVIDIA GPU donation grant. We thank Prof. Hongjing Lu from the UCLA Psychology Department for useful discussions on the motivation of this work, and three anonymous reviewers for their constructive comments.

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
