[Supplementary Material]

# Supplementary Material for Cooperative Holistic Scene Understanding: Unifying 3D Object, Layout, and Camera Pose Estimation

**Siyuan Huang** [1]
huangsiyuan@ucla.edu

**Siyuan Qi** [2]
syqi@cs.ucla.edu

**Yinxue Xiao** [2]
yinxuex@ucla.edu

**Yixin Zhu** [1]
yixin.zhu@ucla.edu

**Ying Nian Wu** [1]
ywu@stat.ucla.edu

**Song-Chun Zhu** [1,2]
sczhu@stat.ucla.edu

[1] Dept. of Statistics, UCLA    [2] Dept. of Computer Science, UCLA

## 3D Object Detection

We report results of 3D object detection of 30 object categories in Table 1.

Table 1: Comparisons of 3D object detection on SUN RGB-D.

| toilet | recycle_bin | night_stand | endtable | drawer | computer | keyboard | table | chair | monitor | stool |
|---|---|---|---|---|---|---|---|---|---|---|
| 58.55 | 10.19 | 6.34 | 8.88 | 4.39 | 0.83 | 0.58 | 26.21 | 17.12 | 0.26 | 6.25 |
| lamp | dresser | picture | garbage_bin | shelf | sofa_chair | cabinet | sink | desk | bookshelf | coffee_table |
| 1.75 | 4.28 | 0 | 7.54 | 3.01 | 32.34 | 2.63 | 5.34 | 9.55 | 2.37 | 11.42 |
| box | sofa | whiteboard | bed | cpu | paper | painting | pillow | mAP | | |
| 1.24 | 41.22 | 0.38 | 63.58 | 3.70 | 0 | 0 | 2.24 | 11.07 | | |

## 3D Box Estimation

We report results of 3D IoU and 2D IoU of 30 object categories in Table 2.

Table 2: 3D box estimation results on SUN RGB-D.

| | toilet | recycle_bin | night_stand | endtable | drawer | computer | keyboard | table | chair | monitor |
|---|---|---|---|---|---|---|---|---|---|---|
| IoU (3D) | 25.08 | 13.17 | 12.50 | 16.24 | 15.78 | 5.21 | 2.89 | 20.77 | 15.66 | 5.36 |
| IoU (2D) | 72.53 | 69.69 | 69.38 | 74.58 | 64.13 | 67.80 | 58.20 | 71.20 | 68.10 | 60.84 |
| stool | lamp | dresser | picture | garbage_bin | shelf | sofa_chair | cabinet | sink | desk | bookshelf |
| 9.17 | 5.88 | 9.29 | 1.03 | 10.13 | 6.90 | 24.03 | 10.07 | 9.87 | 15.64 | 7.10 |
| 74.52 | 63.80 | 69.28 | 51.96 | 67.66 | 62.14 | 75.05 | 69.09 | 59.34 | 70.10 | 59.22 |
| coffee_table | box | sofa | whiteboard | bed | cpu | paper | painting | pillow | mIoU | |
| 23.82 | 4.88 | 27.92 | 2.36 | 33.12 | 8.99 | 0.09 | 0.47 | 5.18 | 11.62 | |
| 75.27 | 60.85 | 74.45 | 63.20 | 75.74 | 65.09 | 60.54 | 61.71 | 62.98 | 66.6 | |

## More Results