[Reviews · NeurIPS 2018]

Reviewer 1



An approach for joint estimation of 3D Layout, 3D Object Detection, Camera Pose Estimation and `Holistic Scene Understanding’ (as defined in Song et al. (2015)) is proposed. More specifically, deep nets, functional mappings (e.g., projections from 3D to 2D points) and loss functions are combined to obtain a holistic interpretation of a scene illustrated in a single RGB image. The proposed approach is shown to outperform 3DGP (Choi et al. (2013)) and IM2CAD (Izadinia et al. (2017)) on the SUN RGB-D dataset. Review Summary: The paper is well written and presents an intuitive approach which is illustrated to work well when compared to two baselines. For some of the tasks, e.g., 3D Layout estimation, stronger baselines exist and as a reviewer/reader I can’t assess how the proposed approach compares. Moreover, a fair comparison to IM2CAD wasn’t possible because of a different dataset split, leaving a single baseline from 2013 (5 years ago). In addition, I think many important aspects are omitted which hinders reproducibility. The latter two reasons (comparison to stronger baselines and dropping important aspects) are the reason for my rating. Quality: The paper is well written. Clarity: Important aspects are omitted which prevent reproducibility according to my opinion. Originality: Could be improved by focusing on omitted aspects. Significance: Hard to judge given a small number of baselines. Details: 1) Why is there an offset \Delta^I between the 2D box center an the projection of the 3D box center in Eq. (1)? This doesn’t seem intuitive. Can the authors explain? 2) I think the parameterization of the room layout remains unclear. Many papers in this direction exist and only three lines are spent to explain how eight 3D points are computed from `center, size, and orientation.’ I don’t think I could reproduce those results based on this short description. 3) Very little is mentioned about the two employed deep nets L_GGN and L_LON. E.g., are they regression losses or classification losses? How are they composed? The paper states `we adopt a hybrid of classification and regression for predicting the sizes and heading angles’ (l.135) but doesn’t provide any more information. I understand that space in the main paper is limited, but the supplementary material is a good place to discuss those aspects. 4) A fair comparison to IM2CAD wasn’t conducted because IM2CAD manually selected a subset of images from the dataset. I’m wondering whether the authors reached out to Izadinia et al. to obtain the list for the test set? A fair comparison would be more compelling such that a reader/reviewer isn’t left with a single baseline from 2013. 5) Beyond a fair comparison to IM2CAD, additional state-of-the-art baselines exist for some of the proposed tasks. E.g., RoomNet (C.-Y. Lee et al., `RoomNet: End-To-End Room Layout Estimation,’ ICCV 2017) and references therein for 3D layout estimation. Comparison to those would make the paper much more compelling since a reader is able to better assess the strength of the proposed approach on individual tasks. Note that there is no need for state-of-the-art results on all tasks, but I would want to know the differences between joint learning and specifically designed deep nets.

Reviewer 2



I have read the rebuttal and I am satisfied with the authors responses. I am happy accepting the paper. ----------- This paper proposes to take an image as input and output: (a) camera pose (b) room layout as a 3D box, and (c) for every object, the 3D box. The method consists of a global network that predicts (a,b) and a local network to predict (c). In addition to the standard losses, the paper proposes physical losses that entangle the global and local predictions by using the predicted camera pose in the 3D bounding box loss and a 2D projected bounding box, as well as a loss that ensures that each object is inside the room. I like many aspects of the submission but I am concerned that the experiments do not clearly support the claims of the paper. + The parametrization of the 3D box is clever and, while it is simple geometry, I do not think I have seen it before. It quite cleanly refactors the estimation of the coordinates so that it's easy to actually get the box to align with the 2D box. + Forcing the two networks to work together makes a lot of sense and the particular formulation of how the two should work together is clean: factoring it so that there's only one camera pose so that the objects are not independently implicitly predicting a camera pose is quite nice. Similarly, the non-intersection loss is quite nice. Experimentally, the paper is tricky to evaluate -- there isn't a lot of work on this task, so it's difficult to do meaningful comparisons. 3DGP and Hedau et al.'s VP estimator are very much out of date, leaving IM2CAD and ablations. My overall concern with the experiments is that I'm not convinced the paper clearly demonstrates the benefits of the system. -It's not clear that the "individual" vs "cooperative" result is explained by the cooperative estimation: estimating 3D pose from a cropped 2D bounding box is a difficult problem, and so it's not surprising that the setting that can see the full image outperforms the network that can only see cropped bounding boxes. Indeed this was observed in Tulsiani et al. CVPR 2018, in which not providing the context of the full image for estimating object pose led to performance losses. -The comparison to IM2CAD is difficult to assess. It's clear that the paper improves on it by a little bit in terms of holistic understanding and by a lot in terms of the layout estimation, but it's unclear where that performance gain comes from. The base architectures change from VGG16 -> Resnet34 for object detection (and maybe also the object detection pipeline) and VGG16 + hand-crafted optimization -> Direct estimation via resnet34 for layout estimation. -The ablative analysis does not clearly support the importance of most of the proposed constraints. For example, S3 ablation (without L_PHY), the drop without L_PHY is not particularly big, mainly <0.5% and sometimes not including L_PHY improves things (precision, for instance). This could easily be explained by plenty of other effects like the natural variance of training runs. Similarly, in the S2 ablation, getting rid of the 2D projection loss improves things dramatically. Smaller stuff that needs to be fixed but which does not impact my review a lot: -The very first sentence is misciting Oliva 2005. Oliva 2005 has nothing to do with getting full, metric 3D out of an image and indeed, comments "Common to these representations is their holistic nature: the structure of the scene is inferred, with no need to represent the shape or meaning of the objects". -SoftNMS is not a detector. SoftNMS is a post-detection heuristic used to improve the output of a detector. The paper should identify which detector was used. - Missing citations. The paper is missing an important line of work from the 2010-2014 era of joint optimization of objects and layout, e.g., Lee, et al. NIPS 2010; Schwing et al. Box in the Box ICCV 2013. (which is notably not slow) -Paragraph discussing the ablative analysis S1/S2/S3 has the order of S2/S3 flipped.

Reviewer 3



This paper aims to infer a 3D representation of a scene in terms of a 3D bounding box per (detected) object (represented via rotation+size+translation), the scene layout (box-like extent), and the camera pose w.r.t. the scene. In addition to independent (regression) losses for each of the predictions (typically what previous methods rely on), this paper also leverages some joint 'cooperative' losses for the training. In particular, a) the per-object rotation/translation/size predictions should yield the correct 3d box, b) the 3D box when projected via the estimated pose should project to the known 2D box, c) the 3D boxes should not lie outside the layout. The experiments indicate the benefits of these additional losses and the paper shows improvements over previous methods for both, layout estimation and 3d object detection. This is a generally well written paper. The primary contributions of the work are the additional joint losses that tie together the global and per-object predictions. I feel that these are simple but novel additions to various existing approaches that typically use individual terms, and that the experiments justify the benefits of these additions. Further, I feel the parametrization of the translation of an object is also a clever detail (though similar to concurrent work '3D-RCNN' by Kundu et. al, CVPR 18). An additional interesting aspect is that the approach can allow learning (not as good, but still meaningful) 3D box prediction using only 2D supervision (S4 in Table 4). I wish the paper provided more visualizations and emphasis for this setting. One aspect I'm not sure about is the difference between the 'world frame' and the camera frame. I'm assuming that the 'world frame' is some axis-aligned frame, where the room is along manhattan directions, and the 'camera pose' In Fig 2 Is w.r.t to this. I am not sure why the layout etc. cannot just be predicted in the camera frame - this would remove the need to infer a global camera pose at all. Additionally, while I generally like the paper, I am hesitant to give a strong acceptance rating given the paper's contributions. While the additional losses are neat and might be applicable to other general scene 3D prediction work, they are arguably obvious additional terms, and I'm also not sure if these contributions are best suited for a learning conference.